# λ-GRPO: UNIFYING THE GRPO FRAMEWORKS WITH LEARNABLE TOKEN PREFERENCES

## ABSTRACT

Reinforcement Learning with Human Feedback (RLHF) has been the dominant approach for improving the reasoning capabilities of Large Language Models (LLMs). Recently, Reinforcement Learning with Verifiable Rewards (RLVR) has simplified this paradigm by replacing the reward and value models with rule-based verifiers. A prominent example is Group Relative Policy Optimization (GRPO). However, GRPO inherently suffers from a *length bias*, since the same advantage is uniformly assigned to all tokens of a response. As a result, longer responses distribute the reward over more tokens and thus contribute disproportionately to gradient updates. Several variants, such as DAPO and Dr. GRPO, modify the token-level aggregation of the loss, yet these methods remain heuristic and offer limited interpretability regarding their implicit token preferences. In this work, we explore the possibility of allowing the model to *learn its own token preference* during optimization. We unify existing frameworks under a single formulation and introduce a learnable parameter $\lambda$ that adaptively controls token-level weighting. We use λ-GRPO to denote our method, and we find that λ-GRPO achieves consistent improvements over vanilla GRPO and DAPO on multiple mathematical reasoning benchmarks. On *Qwen2.5* models with 1.5B, 3B, and 7B parameters, λ-GRPO improves average accuracy by $+1.9\%$, $+1.0\%$, and $+1.7\%$ compared to GRPO, respectively. Importantly, these gains come without any modifications to the training data or additional computational cost, highlighting the effectiveness and practicality of learning token preferences. Our code is available at: https://anonymous.4open.science/r/Lambda-GRPO-AD74/.

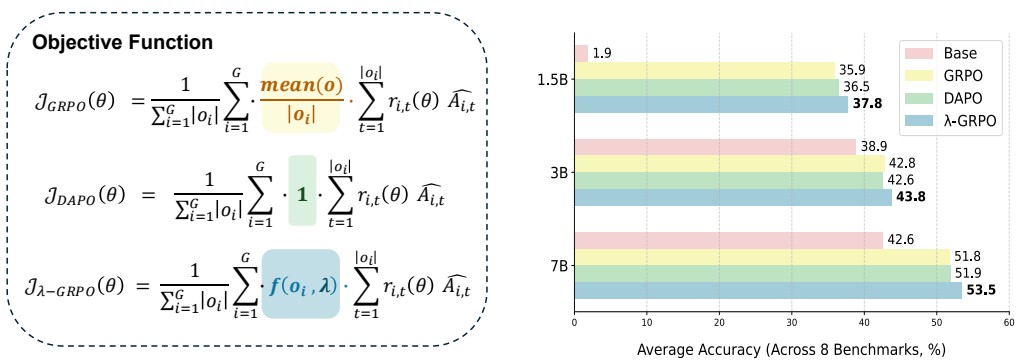

Figure 1: The key distinction among GRPO, DAPO, and λ-GRPO lies in their token-aggregation schemes (highlighted in the shaded region). Across *Qwen2.5* models of multiple sizes (1.5B, 3B, and 7B) and 8 benchmarks, λ-GRPO consistently outperforms others, with a learnable parameter $\lambda$.

## 1 INTRODUCTION

Large language models (LLMs) with o1-like systems (OpenAI, 2024; Wu et al., 2024; DeepSeek-AI, 2025) have been widely applied to complex reasoning tasks such as mathematical problem solving.

A common approach to training long chain-of-thought reasoning is Reinforcement Learning with Human Feedback (RLHF) (Xu et al., 2025), where models explore trajectories and are optimized toward preferred outcomes. The most popular optimization method in this paradigm is Proximal Policy Optimization (PPO) (Schulman et al., 2017). However, PPO training is often unstable and computationally expensive (Zheng et al., 2023; Wu et al., 2023). To address these limitations, Group Relative Policy Optimization (GRPO) (Shao et al., 2024) removes the need for a separate value model and reward model, instead relying on a rule-based verifier as the training signal. This design has proven highly effective in domains requiring verifiable reasoning, spanning mathematical problem solving (DeepSeek-AI, 2025), code programming (Liu & Zhang, 2025), search agent (Jin et al., 2025) and even multimodal reasoning (Huang et al., 2025).

Despite these advances, modern policy and preference optimization methods for post-training almost always exhibit a strong *length bias*, the tendency for models to generate unnecessarily long responses even when concise answers would suffice. For example, Singhal et al. (2024) showed that much of the reward gains in RLHF stem from response length increases rather than substantive improvements in quality. Similarly, Hu et al. (2025) found that reward models and preference datasets themselves often favor longer responses, which leads trained policies to exploit this bias by producing verbose outputs. This issue also persists in GRPO, since the advantage is applied uniformly across all tokens of a response (Shao et al., 2024). Recent variants address this bias by modifying how token rewards are aggregated. DAPO (Yu et al., 2025) enforces uniform token-level aggregation so that all tokens across responses share equal weight, while Dr. GRPO (Liu et al., 2025) aggregates token-level gradients without variance normalization. In other words, GRPO effectively lets tokens in a long response share one reward, DAPO spreads weight evenly across all tokens of all responses, and Dr. GRPO further adjusts a modification to the advantage function. Although these strategies partially mitigate length bias, they remain heuristic and lack the flexibility to adapt across tasks. These observations raise a critical question:

*Can we let the model itself decide its token preference?*

In other words, rather than hard-coding whether long or short replies should be favored, can the optimization process learn a tunable, context-aware preference over response length?

To this end, we propose a unified framework, denoted as $\lambda$-GRPO, for token-level preference optimization with a *learnable weighting scheme*. Unlike GRPO, DAPO, and Dr. GRPO that rely on heuristics, our method introduces a parametric weighting function with a tunable parameter $\lambda$ that adapts dynamically to the distribution of response lengths. This design enables the model to determine its own token preference in a context-aware manner. We summarize the main difference in Figure 1, and highlight how existing methods arise as special cases of our formulation. Empirically, we demonstrate that $\lambda$-GRPO consistently achieves stronger performance across *Qwen2.5* models of multiple scales (1.5B, 3B, and 7B) on 8 reasoning benchmarks. Our main contributions are summarized as follows:

- We propose **a unified formulation** that encompasses GRPO, DAPO, and Dr. GRPO as special instances of token-level preference optimization. This approach clarifies the implicit assumptions each method makes regarding length aggregation.

- We introduce **a learnable preference framework** that adaptively reweights token contributions based on response length distributions, enabling the model to learn its own preference.

- $\lambda$-GRPO is empirically validated across reasoning benchmarks, demonstrating improved performance, reduced verbosity, and enhanced training stability compared to prior-arts.

## 2 METHODOLOGY

### 2.1 PRELIMINARY

**Reinforcement Learning with Verifiable Rewards (RLVR)** views text generation with an LLM as a sequential decision process. At step $t$, the state is the prompt $x$ together with the partial output $o_{1:t-1}$, and the action is to produce the next token $o_t \sim \pi_\theta(\cdot \mid x, o_{1:t-1})$. The policy $\pi_\theta$ denotes the current model we are optimizing, while $\pi_{\theta_{\text{old}}}$ is the frozen policy from the previous iteration that is used to sample trajectories for stable training. The reward function $r(x, o)$ is defined by

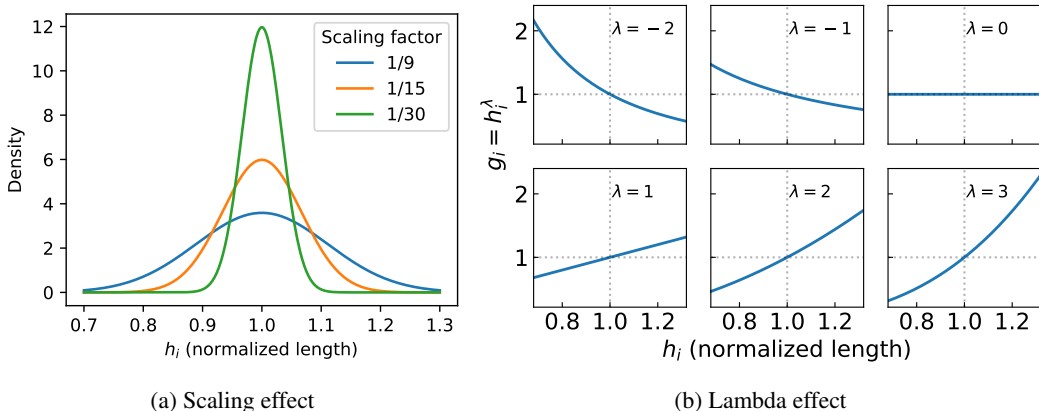

(a) Scaling effect                                  (b) Lambda effect

Figure 2: Illustration of our learnable preference design. (a) shows how the scaling factor $1/k$ influences the distribution of normalized lengths $h$. Larger scaling values produce a wider spread, magnifying differences between long and short responses, smaller values compress the distribution around 1, reducing sensitivity to length deviations. (b) shows how the exponent $\lambda$ adjusts the direction of token preference through $g_i = h_i^\lambda$. When $\lambda = 0$, all responses are weighted equally; positive $\lambda$ favors longer responses, negative $\lambda$ emphasizes shorter ones.

deterministic rules (e.g., correctness of an answer or validity of code), ensuring that the feedback signal is verifiable rather than subjective. The learning objective combines two ingredients: a policy gradient loss that increases the likelihood of outputs with higher advantage values, and a KL penalty that keeps the updated policy not far from a fixed reference distribution $\pi_{\theta_{\mathrm{ref}}}$. The resulting objective is:

$$\mathcal{J}_{\mathrm{RLVR}}(\theta) = \mathbb{E}_{x \sim \mathcal{D}, \, o \sim \pi_{\theta_{\mathrm{old}}}(\cdot|x)}\big[r_{i,t}(\theta)\, A(x, o)\big] + \beta\, \mathbb{E}_{x \sim \mathcal{D}}\big[\mathrm{KL}\big(\pi_{\theta_{\mathrm{ref}}}(\cdot|x) \, \| \, \pi_\theta(\cdot|x)\big)\big], \quad (1)$$

where $A(x, o)$ is the advantage function that measures how much better a completion $o$ performs. The KL term is optional, with a coefficient $\beta$ that controls the trade-off between maximizing the verifiable reward and staying close to the reference model. $r$ is the importance sampling ratio comparing the new and old policy:

$$r_{i,t}(\theta) = \frac{\pi_\theta(o_{i,t} \mid x, o_{i,<t})}{\pi_{\theta_{\mathrm{old}}}(o_{i,t} \mid x, o_{i,<t})} \quad (2)$$

**GRPO** was proposed as a lightweight alternative to PPO that avoids learning an additional value function while still providing a stable training signal (Shao et al., 2024). The key idea is to measure the quality of each sampled output relative to a group of alternatives drawn from the same policy. Rather than fitting a value baseline, GRPO defines the advantage of $o_i$ by subtracting the average reward of the group and scaling by its standard deviation:

$$\hat{A}_{i,t} = \frac{R(x, o_i) - \mathrm{mean}(\{R(x, o_j)\}_{j=1}^G)}{\mathrm{std}(\{R(x, o_j)\}_{j=1}^G)} \quad (3)$$

This group-normalized advantage is then assigned uniformly to all tokens in the response, leading to the clipped surrogate loss

$$\mathcal{J}_{\mathrm{GRPO}}(\theta) = \frac{1}{G} \sum_{i=1}^G \frac{1}{|o_i|} \sum_{t=1}^{|o_i|} \min\Big(r_{i,t}(\theta)\,\hat{A}_{i,t},\, \mathrm{clip}\big(r_{i,t}(\theta),\, 1-\epsilon,\, 1+\epsilon\big)\hat{A}_{i,t}\Big) \quad (4)$$

where $G$ denotes the number of rollouts (i.e., sampled responses per prompt), $|o_i|$ is the length of the $i$-th response.

**DAPO** A central feature of DAPO is that the policy gradient is applied directly at the token level rather than treating the entire response as a single unit. This design ensures that every token within

a sampled output $o_i$ contributes proportionally to the gradient update. This fine-grained token-level optimization improves stability and provides more informative feedback for training LLMs.

Formally, the DAPO objective is defined as

$$\mathcal{J}_{\text{DAPO}}(\theta) = \frac{1}{\sum_{i=1}^{G} |o_i|} \sum_{i=1}^{G} \sum_{t=1}^{|o_i|} \min\Big( r_{i,t}(\theta)\, \hat{A}_{i,t},\ \text{clip}\big(r_{i,t}(\theta),\, 1-\epsilon,\, 1+\epsilon'\big)\, \hat{A}_{i,t} \Big) \quad (5)$$

**Dr. GRPO** (Liu et al., 2025) was proposed to address systematic biases in the original GRPO formulation. In GRPO, the advantage of each response is normalized by the group's standard deviation, and the policy loss is further averaged across response length.

$$\mathcal{J}_{\text{Dr.GRPO}}(\theta) = \frac{1}{G} \sum_{i=1}^{G} \sum_{t=1}^{|o_i|} \min\Big( r_{i,t}(\theta)\, \hat{A}_{i,t},\ \text{clip}\big(r_{i,t}(\theta),\, 1-\epsilon,\, 1+\epsilon\big)\, \hat{A}_{i,t} \Big) \quad (6)$$

## 2.2 UNIFIED TOKEN PREFERENCE

The policy optimization objectives of GRPO, DAPO, and Dr. GRPO can be expressed under a unified framework that we call *Unified Token Preference*. The key observation is that all three methods share the same token-level clipped surrogate structure, but differ in the weighting function $f(o_i)$ applied to each sampled output $o_i$. Formally, the unified objective can be written as

$$\mathcal{J}(\theta) = \frac{1}{\sum_{i=1}^{G} |o_i|} \sum_{i=1}^{G} f(o_i) \sum_{t=1}^{|o_i|} \min\Big( r_{i,t}(\theta)\, \hat{A}_{i,t},\ \text{clip}\big(r_{i,t}(\theta),\, 1-\epsilon,\, 1+\epsilon\big)\, \hat{A}_{i,t} \Big) \quad (7)$$

The weighting function $f(o_i)$ specifies how each output contributes to the token-level aggregation:

$$f_{\text{GRPO}}(o_i) = \frac{\mu}{|o_i|} \quad (8)$$

$$f_{\text{DAPO}}(o_i) = 1 \quad (9)$$

$$f_{\text{Dr.GRPO}}(o_i) = \mu \quad (10)$$

where $\mu$ denotes the average response length across the group, i.e., $\mu = \frac{1}{G} \sum_{i=1}^{G} |o_i|$.

This formulation highlights that GRPO effectively downweights long responses, DAPO treats all responses equally at the token level, and Dr. GRPO applies a uniform length-based scaling across responses. By casting them into the same token preference framework, we can analyze their differences in terms of how they balance output length and reward signal during optimization.

## 2.3 OUR LEARNABLE PREFERENCE

GRPO, DAPO, and Dr. GRPO can all be expressed in the unified token preference framework by specifying different fixed weighting functions $f(o_i)$. However, they remain heuristic in nature and cannot adapt to the distribution of response lengths. In particular, GRPO implicitly possesses token-weighting bias, DAPO enforces uniform token-level normalization, and Dr. GRPO applies a global scaling based on average length. These designs do not allow the optimization process to flexibly balance between long and short responses across different prompts.

To overcome this limitation, we propose a *learnable preference* that treats response length as a stochastic variable and derives adaptive weights through a normalized transformation. Given $G$ responses $\{o_i\}_{i=1}^{G}$, we first compute their mean and variance

$$\mu = \text{mean}(o) \quad (11)$$

$$\sigma = \text{std}(o) \quad (12)$$

And then standardize the length:

$$z_i = \frac{o_i - \mu}{\sigma} \quad (13)$$

$$h_i = 1 + r\, z_i \quad (14)$$

$r > 0$ is a scaling factor. This construction ensures that $h_i$ is centered around 1, with deviations controlled by $r$. Illustratively, about 99.7% of $z_i = \frac{o_i - \mu}{\sigma}$ lie within $[-3, 3]$, implies that $h_i \in [1 - 3r, 1 + 3r]$ for almost all responses. As shown in Figure 2a, this corresponds to $h_i \in [0.9, 1.1]$ for $r = \frac{1}{30}$ (conservative), $h_i \in [0.8, 1.2]$ for $r = \frac{1}{15}$ (moderate), and $h_i \in [\frac{2}{3}, \frac{4}{3}]$ for $r = \frac{1}{9}$ (aggressive).

We then apply an exponent $\lambda$ to adjust the relative contribution:

$$g_i = h_i^\lambda \tag{15}$$

**On the role of $\lambda$** The exponent $\lambda$ in $g_i = h_i^\lambda$ controls how response length affects the weighting function, and can be interpreted as a **tunable** token preference. Figure 2b illustrates this effect. When $\lambda = 0$, all $g_i = 1$, meaning that every response is treated equally regardless of its normalized length $h_i$. When $\lambda > 0$, $g_i$ grows with $h_i$, which emphasizes longer responses and downweights shorter ones. Conversely, when $\lambda < 0$, $g_i$ decreases with $h_i$, assigning more weight to shorter responses while suppressing long ones. Thus, $\lambda$ serves as a direct control knob: positive values bias the optimization toward long outputs, negative values favor short outputs, and $\lambda = 0$ corresponds to length-neutral weighting. This flexibility allows our method to adaptively model token preferences, rather than fixing them by hand as in prior approaches. Overall, $\lambda$ sets the *direction* of token preference, and scaling factor $r$ sets its *strength*, offering an interpretable knob to trade off verbosity vs. brevity without distorting the expectation across methods.

The scale of $g$ can vary significantly across groups and across different $\lambda$ values. To stabilize training and make weights comparable, we normalize $g$ with a softmax:

$$f(o) = \text{softmax}(g) \times G \tag{16}$$

The softmax ensures that the weights depend only on the relative differences among $g_i$, preventing abnormally large values from dominating the gradient. Moreover, rescaling by $G$ keeps the total weight consistent with the unified token preference framework, ensuring fair comparison with GRPO, DAPO, and Dr. GRPO. This design makes the training dynamics more controllable: regardless of the absolute scale of $g$, the resulting $f(o_i)$ always sums to $G$ within each group. The detailed derivation of the gradient with respect to $\lambda$ is provided in Appendix C, offering further mathematical insight into how the coefficient is updated during training.

**Implementation Detail of $\lambda$** In our proposed method, we introduce a learnable coefficient $\lambda$ that is optimized jointly with the model parameters. Specifically, $\lambda$ is initialized as a trainable tensor on the GPU, and we use stochastic gradient descent (SGD) to optimize it. The coefficient $\lambda$ is initialized with a value 0) and is treated as a learnable parameter during training. During training, gradients for $\lambda$ are accumulated together with the overall loss, and $\lambda$ is updated at each optimization step.

The initialization and optimization process are as follows:

---

**$\lambda$ Parameter Optimizer**

```
lambda = torch.nn.Parameter(
    torch.tensor([val], dtype=torch.float32, device= cuda ))
lambda_opt = torch.optim.SGD(
    [{ params : [lambda],  lr : 1e-1,  weight_decay : 0.0}])
```

---

## 2.4 WHY IT WORKS

We cast GRPO/DAPO/Dr. GRPO into a unified objective where all differences reduce to the sample-level aggregation weight $f(o)$. We replace fixed heuristics with a data-adaptive weighting: standardize response lengths within the group. The sign of $\lambda$ determines whether training tilts toward longer or shorter outputs (neutral at $\lambda = 0$), while $r$ controls sensitivity; the groupwise softmax preserves total weight and stabilizes aggregation. Consequently, the policy learns how much gradient budget each response should receive as a function of the observed length–reward relationship. This yields a lower-variance, better-calibrated optimization signal than fixed heuristics, and adapts when verbosity helps (or hurts) verifiable reward.

## 3 EXPERIMENTAL SETUP

**Training Datasets**   We follow the SimpleRL Zoo setting  (Zeng et al., 2025), adopting GSM8K (Cobbe et al., 2021) and MATH  (Hendrycks et al., 2021) training datasets. In line with SimpleRL Zoo's setup, we emphasize that the difficulty of training data is crucial for the success of zero RL, where training may start directly from the base models in a way that accords with the base models' natural ability to explore and discover rewardable behaviors, i.e., the capacity to sample diverse reasoning trajectories and uncover rewardable behaviors without much guidance. The data is partitioned into three difficulty levels: Easy (GSM8K and MATH level 1), Medium (MATH levels 1–4), and Hard (MATH levels 3–5). We use the hard level with $8,523$ problems as our training set, and *qwen-boxed* as our chat template, with an example in Appendix D.

**Evaluation**   We evaluate the performance of the fine-tuned models based on the SimpleRL  (Zeng et al., 2025) framework, using $8$ widely used math reasoning benchmarks: GSM8K  (Cobbe et al., 2021), MATH500  (Hendrycks et al., 2021), Minerva  (Lewkowycz et al., 2022), Gaokao (Zhang et al., 2023), OlympiadBench (He et al., 2024b), and College Math  (Innovations, 2024), AIME24 (LI et al., 2024), and AMC23 (He et al., 2024a), relying on the Qwen Math codebase that was also used in  Zeng et al. (2025). This setup ensures consistency with prior work and allows reproducible evaluation across multiple math reasoning benchmarks. For AMC and AIME, we report Average@32: for each problem, we generate 32 independent rollouts (stochastic samples) and compute the mean Pass@1 accuracy across these 32 attempts; the final score is the average of these per-item means over the dataset. For the remaining six benchmarks, we report the standard Pass@1 using a single prediction per problem. The statistics of these benchmarks are listed in Appendix F.

**Detail Implementation**   Our training is based on the VeRL framework, implementing two provided token weighting schemes:  sequence-mean-token-mean (vanilla GRPO) and token-mean (DAPO-like GRPO). We also implement our customized weighting scheme and enable the learnable lambda. We use a rule-based reward, adding a format penalty on the basic correctness (ground truth) that assigns $+1$ for correct answers, $-0.5$ for incorrect answers with correct boxed formats, and $-1$ for incorrect answers with incorrect formats. We fine-tune the base models *Qwen2.5-1.5B*, *Qwen2.5-3B*, and *Qwen2.5-7B* for two epochs (160 steps) using $4\times$ NVIDIA A800-SXM4-80GB GPUs for the 1.5B/3B models and $8\times$ NVIDIA A800-SXM4-80GB GPUs for the 7B model, with a batch size of $1024$, mini batch size of $256$, micro batch size of $8$, a max response length of $2048$, and a reducer ($r$) value of $\frac{1}{9}$. For $\lambda$ updating, we use stochastic gradient descent with a relatively large learning rate $1e-1$ and no weight decay.

**Baselines**   We compare our method with vanilla GRPO and GRPO with DAPO-like token-mean weighting variant. For simplicity, we directly refer to the latter as "DAPO" in the results. We do not include Dr. GRPO as a separate baseline. Our comparison focuses on token-level loss aggregation. From this angle, Dr. GRPO differs from DAPO only by multiplying a group-constant factor (the rollout group's mean token length, according to Eq. 2.2), and thus does not constitute a distinct aggregation rule.  To guarantee a fair comparison, we train all methods under identical datasets, prompt templates, and model architectures.

## 4 EVALUATION

### 4.1 BENCHMARK PERFORMANCE

> **Key Findings**
>
> 1. $\lambda$-GRPO consistently outperforms baselines (DAPO and GRPO) on mathematical reasoning tasks across model scales from 1.5B to 7B parameters.
> 2. Its performance gains are more significant on smaller models and more challenging benchmarks.

Table 1 reports the performance of *Qwen2.5* base models across math reasoning tasks. As expected, the base models without RL are extremely weak, with average scores of 1.9, 38.9, and 42.6 for the 1.5B, 3B, and 7B variants, respectively. This highlights the necessity of reinforcement learning in stimulating the reasoning potential of these models.

Table 1: Performance on math reasoning using Qwen2.5 models.

| Model | GSM8K | MATH500 | Minerva | Gaokao | Olympiad | College Math | AIME | AMC | Avg. |
|---|---|---|---|---|---|---|---|---|---|
| | | | | Qwen2.5-1.5B | | | | | |
| **Base** | 8.5 | 1.8 | 1.8 | 1.8 | 1.8 | 1.3 | 0 | 0.3 | 1.9 |
| DAPO | 75.7 | 58.0 | 20.2 | 50.4 | 20.6 | 33.0 | 3.3 | 31.5 | 36.5 |
| GRPO | 75.1 | 57.0 | **21.7** | 49.6 | 20.3 | 32.4 | **4.2** | 26.9 | 35.9 |
| $\lambda-$GRPO | **77.3** | **59.2** | 21.0 | **50.6** | **21.3** | **35.0** | 3.4 | **34.7** | **37.8** |
| | | | | Qwen2.5-3B | | | | | |
| **Base** | 81.6 | 58.8 | 25.0 | 47.0 | 24.1 | 32.8 | 6.5 | 35.8 | 38.9 |
| DAPO | 85.1 | 65.4 | 31.2 | 53.0 | 27.6 | 36.6 | 6.2 | 35.9 | 42.6 |
| GRPO | **85.7** | 64.0 | 31.2 | 53.0 | **27.9** | 37.0 | **7.7** | 36.5 | 42.8 |
| $\lambda-$GRPO | 85.3 | **67.4** | **32.7** | **54.0** | 27.7 | **37.2** | 7.3 | **38.7** | **43.8** |
| | | | | Qwen2.5-7B | | | | | |
| **Base** | 87.8 | 63.6 | 26.8 | 53.5 | 29.9 | 34.6 | 8.0 | 36.6 | 42.6 |
| DAPO | 91.0 | **77.8** | 36.4 | 63.9 | 37.9 | 40.4 | 14.2 | 54.1 | 51.9 |
| GRPO | 89.7 | 77.2 | **40.1** | 62.6 | 35.6 | 38.6 | **17.2** | 52.6 | 51.8 |
| $\lambda-$GRPO | **92.0** | 75.8 | **40.1** | **66.2** | **38.1** | **41.4** | 15.1 | **58.8** | **53.5** |

Across model scales, our method consistently outperforms both GRPO and DAPO. On *Qwen2.5-1.5B*, it attains an average score of 37.8, exceeding DAPO (+1.3%) and GRPO (+1.9%). This is a notable improvement on smaller models like 1.5B, with standard training already near saturation. These results suggest that our dynamic weighting scheme is especially beneficial for less capable base models, providing stronger guidance than standard group-wise optimization. On the *Qwen2.5-3B* base model, our method further improves the average to 43.8, showing consistent gains over both DAPO (+1%) and GRPO (+1.2%). Notably, the improvements are concentrated on challenging tasks such as MATH500, Minerva, and Gaokao, demonstrating that our approach leverages difficult samples very effectively to boost complex reasoning. For the larger *Qwen2.5-7B* model, the base model already exhibits stronger reasoning ability; the advantage of our method remains sufficient, with average results of 53.5 compared to 51.9 on DAPO (+1.6%) and 51.8 on GRPO (+1.7%). This indicates that while self-exploration already benefits strong models, our proposed approach still provides complementary improvements by balancing stability and exploration. Overall, these results confirm that our method consistently improves reasoning performance across scales, with clear gains on smaller models due to a more fragile optimization landscape and the benefit of adaptive weighting, and steady, incremental gains on larger models through a better balance between stability and exploration.

**Analysis across the benchmarks**     To investigate how training progresses under different optimization methods, we plot the accuracy curves across multiple reasoning benchmarks as a function of training steps (Figure 3). Across benchmarks, accuracy mostly increases with training steps and typically plateaus around step 120. *λ-GRPO* is consistently the best or tied for the best on GSM8K, MATH500, Gaokao, Olympiad, College Math, AIME@32, and AMC@32. The improvements over *DAPO* are most visible after step 80, e.g., College Math, AIME, and AMC, while *GRPO* is competitive early on but generally lags by the end of training. These results indicate that, on *Qwen2.5-1.5B*, our method yields steadier gains and stronger final accuracy without requiring additional schedule tuning. We also include the training curves for *Qwen2.5-3B* in Appendix E.

## 4.2    LEARNING DYNAMICS

> **Key Findings**
>
> 1. λ-GRPO maintains a consistently higher token-level entropy than DAPO, indicating enhanced response diversity and training stability.
> 2. This higher diversity is achieved without increasing response length, demonstrating a more efficient exploration-exploitation balance.

**Entropy**     Entropy generally indicates the diversity and freedom of a generated response. Conditional on correctness, a higher-entropy answer suggests that the model exhibits both robust, correct reasoning and the ability to employ a variety of methods in the solving process. Thus, we encourage

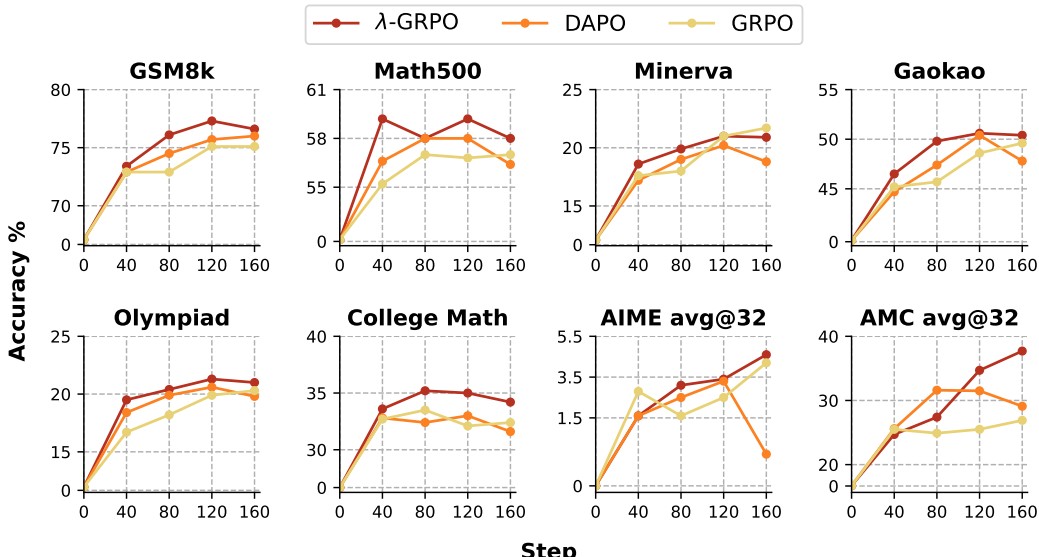

Figure 3: Training curves for the *Qwen2.5-1.5B* base model across benchmarks. Each panel shows accuracy (%) versus training step for three methods (*λ-GRPO*, *DAPO*, and *GRPO*). Steps are {0, 40, 80, 120, 160}.

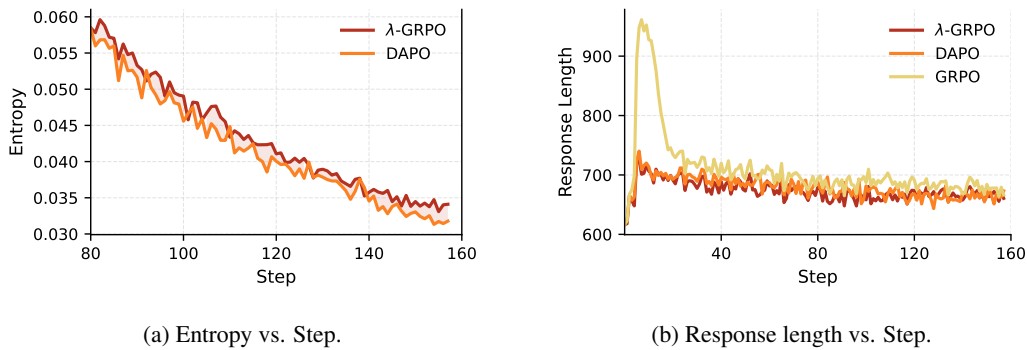

(a) Entropy vs. Step.

(b) Response length vs. Step.

Figure 4: Comparison of our method against GRPO and DAPO across two diagnostics: (a) entropy and (b) response length. Both experiments are based on *Qwen2.5-3B*.

models with higher entropy. Previous studies have reported entropy collapse in RLHF (Cui et al., 2025; Kirk et al., 2023), making entropy a particularly informative metric to monitor. To analyze average token-level entropy across training, we plot entropy vs. steps, as shown in Figure 4a. As reported by DAPO (Yu et al., 2025), DAPO exhibits higher entropy than GRPO overall. Therefore, in Figure 4a we focus on comparing the entropy of λ-GRPO against DAPO.

Within the zoomed window (steps 80–160), all curves steadily decrease, indicating continued convergence and increasingly confident outputs. The λ-GRPO curve stays consistently above the DAPO curve, and the shaded gap remains roughly constant. At similar convergence levels, λ-GRPO keeps higher output entropy, more diversity and exploration without drifting. This helps prevent early collapse and keeps training stable.

**Response length**  Another notable signal comes from response length, which reflects token-weighting preferences rather than raw capability. We include the response length during training in Figure 4b. All methods show an early spike and then quickly stabilize. GRPO peaks near 950 tokens in the very first steps and then settles to 680–700, indicating a stronger transient tendency to produce longer responses. In contrast, λ-GRPO and DAPO stabilize earlier and remain close

(660–690). Paired with Figure 4a, $\lambda$-GRPO attains the entropy advantage without systematically inflating response length, indicating a better exploration–exploitation balance: more informative diversity at roughly the same or lower verbosity.

## 5 RELATED WORK

**Reinforcement Learning with Verifiable Rewards (RLVR)** Recent progress in reasoning-centric training for LLMs has been driven by the use of verifiable signals, such as correctness in math or code, to replace costly annotations of human preferences. This paradigm, known as RLVR, has given rise to a series of policy optimization methods. A prominent line of work is *Group Relative Policy Optimization* (GRPO), used in DeepSeek-R1-Zero (Shao et al., 2024). GRPO extends Proximal Policy Optimization (PPO) (Schulman et al., 2017) by shifting from token-level advantage estimation with a learned critic to group-based normalization: for each query, multiple sampled responses are generated and compared, and updates are based on their relative rewards. This group-wise structure improves sample efficiency and removes the need for a separate value network.

GRPO has inspired a number of extensions refining how the learning objective is constructed. GMPO (Zhao et al., 2025) stabilizes training by maximizing the geometric rather than arithmetic mean of token-level importance ratios, making the updates less sensitive to outliers. GRPO-CARE (Chen et al., 2025) introduces a two-tier reward: in addition to correctness, it adds a consistency bonus derived from a slowly updated reference model, thereby encouraging coherent reasoning without relying solely on KL regularization. DFPO (Jang et al., 2024) (Degeneration-Free Policy Optimization) modifies PPO-style KL penalties with a masking mechanism that blocks harmful updates, mitigating mode collapse and degeneration. GiGPO (Group-in-group policy optimization) (Feng et al., 2025) further extends the group-crediting idea to long-horizon settings by nesting groups at both episode and step levels, providing critic-free updates while incorporating finer-grained rewards. Finally, KRPO (Wang et al., 2025) replaces the simple batch mean baseline of GRPO with a Kalman filter that adaptively tracks latent reward statistics, yielding more stable advantages. GSPO (Zheng et al., 2025) departs from token-level importance ratios altogether, performing sequence-level clipping and reward assignment, which reduces variance and has been shown effective in scaling to larger or MoE-style models.

**Token-weighting and Normalization Biases** GRPO removes the critic but introduces biases in how token-level losses are aggregated: uniform token advantages let longer responses accrue more gradient regardless of quality, and length-based scaling can reward brevity when correct yet under-penalize verbose errors. These length-related biases have motivated methods that explicitly reweight or normalize token contributions. Two recent approaches illustrate this direction. *Dynamic sAmpling Policy Optimization* (DAPO) (Yu et al., 2025) introduces length-aware sampling weights that rescale token-level advantages, ensuring that both short and long responses contribute fairly and preventing models from drifting toward neither overly short nor verbose completions. Dr. GRPO (Liu et al., 2025) addresses a complementary issue: sensitivity to reward noise. By adopting a robust optimization objective, it reduces the impact of outlier responses and provides more stable updates in noisy or heterogeneous reward settings. Both methods highlight that, beyond the choice of optimization algorithm, controlling how token-level signals are weighted is crucial for stable and length-robust RLVR training. Concurrently, Token Hidden Reward (Deng et al., 2025) reweights per-token updates in GRPO using a "hidden reward," exposing a knob $p$ that steers exploitation versus exploration.

## 6 CONCLUSIONS

In this work, we explored the feasibility of moving beyond manually designed heuristics for token aggregation and instead allowing the model to learn its own token-level preference. We proposed a unified framework that reformulates existing methods such as GRPO, DAPO, and Dr. GRPO, and introduced a tunable parameter $\lambda$ that explicitly encodes token preference. When $\lambda = 0$, the model is length-neutral; when $\lambda > 0$, the optimization favors longer responses; and when $\lambda < 0$, it favors shorter ones. This learnable preference provides a flexible mechanism to balance different response lengths across tasks, reducing the reliance on fixed aggregation strategies. Empirical results on multiple mathematical reasoning benchmarks demonstrate that our method consistently improves over existing GRPO variants without introducing additional data or computational cost.

## 7 ETHICS STATEMENT

All experiments were performed using publicly available datasets and models, with no involvement of human subjects or sensitive data. The authors affirm their compliance with the ICLR Code of Ethics and the principles of scientific integrity.

## 8 REPRODUCIBILITY STATEMENT

The paper specifies the GRPO, DAPO, and $\lambda$-GRPO objectives in Section 2; the training dataset, evaluation benchmarks, and implementation details (key hyperparameters, hardware, and experimental settings) are provided in Section 3 and Appendix F; the chat template is given in Appendix D. Reproducible experimental results appear in Section 4.

As anonymized supplementary materials on OpenReview, we include core code at `https://anonymous.4open.science/r/Lambda-GRPO-AD74/` for (i) training base models with GRPO, DAPO, and our proposed $\lambda$-GRPO, and (ii) constructing and optimizing the learnable parameter $\lambda$. We also provide configuration files (hyperparameters, seeds), run scripts, and environment specifications to facilitate exact reruns. After the anonymity period, we will release the full source code and documentation.

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

## A  LIMITATIONS

First, our experiments focus on `Qwen2.5` base models and do not evaluate cross-family generalization; extending to additional architectures (i.e., Llama) would test robustness and portability. Second, we explore only a small set of reducer values, which control the post-weighting scaling range; exploring a wider range or adaptive schedules of the reducer could potentially improve stability and performance and reveal sensitivity to this hyperparameter.

## B  THE USE OF LLMS

Our use of LLMs was limited to grammar and style editing; we did not employ them for research functions (e.g., idea generation). We supplied the original manuscript to OpenAI GPT-5 and asked it to improve clarity and tone to make our work a more professional research paper. All generated text was subsequently checked for factual correctness and fidelity to the source.

## C  DERIVATION OF THE PROPOSED OBJECTIVE FUNCTION

According to section 2.3, the objective is

$$\mathcal{J}(\lambda - GRPO) = \frac{1}{\sum_{i=1}^{G} |o_i|} \sum_{i=1}^{G} f(o_i) L_i = \frac{G}{\sum_{i=1}^{G} |o_i|} \sum_{i=1}^{G} s_i L_i$$

, where

$$L_i = \sum_{t=1}^{|o_i|} \min\left( r_{i,t}(\theta)\, \hat{A}_{i,t},\; \text{clip}\big(r_{i,t}(\theta),\, 1-\epsilon,\, 1+\epsilon\big)\, \hat{A}_{i,t} \right)$$

Here, we show the detailed gradient derivation with respect to $\lambda$.

First, we want to calculate $\frac{\partial g_i}{\partial \lambda}$, where $g_i = h_i^\lambda$.

We can substitute $h_i^\lambda$ as $e^{\ln h_i^\lambda}$, and

$$\frac{\partial(e^{\ln h_i^\lambda})}{\partial \lambda} = \frac{\partial(e^{\lambda \cdot \ln h_i})}{\partial \lambda} = e^{\ln h_i^\lambda} \cdot \ln h_i$$

Hence,

$$\boxed{\frac{\partial g_i}{\partial \lambda} = \frac{\partial}{\partial \lambda}\big(h_i^\lambda\big) = h_i^\lambda \cdot \ln h_i} \tag{17}$$

Then, we will take softmax derivative $\frac{\partial s_i}{\partial \lambda}$, where $s_i = \frac{e^{g_i}}{\sum_{j=1}^G e^{g_j}}$.

Let

$$s_i = \frac{e^{g_i}}{Z}, \qquad Z = \sum_{j=1}^G e^{g_j}.$$

Then, by the quotient rule,

$$\frac{\partial s_i}{\partial \lambda} = \frac{\partial}{\partial \lambda}\left( \frac{e^{g_i}}{Z} \right)$$

$$= \frac{e^{g_i} \frac{\partial g_i}{\partial \lambda} Z - e^{g_i} \frac{\partial Z}{\partial \lambda}}{Z^2}.$$

Using $s_i = \frac{e^{g_i}}{Z}$, we rewrite

$$\frac{\partial s_i}{\partial \lambda} = s_i \left( \frac{\partial g_i}{\partial \lambda} - \frac{1}{Z} \frac{\partial Z}{\partial \lambda} \right).$$

Moreover,

$$\frac{1}{Z} \frac{\partial Z}{\partial \lambda} = \frac{1}{Z} \sum_{j=1}^G \frac{\partial e^{g_j}}{\partial g_j} \frac{\partial g_j}{\partial \lambda} = \frac{1}{Z} \sum_{j=1}^G e^{g_j} \frac{\partial g_j}{\partial \lambda} = \sum_{j=1}^G \frac{e^{g_j}}{Z} \frac{\partial g_j}{\partial \lambda} = \sum_{j=1}^G s_j \frac{\partial g_j}{\partial \lambda}.$$

Substituting back gives the desired result:

$$\boxed{\frac{\partial s_i}{\partial \lambda} = s_i \left( \frac{\partial g_i}{\partial \lambda} - \sum_{j=1}^G s_j \frac{\partial g_j}{\partial \lambda} \right)} \tag{18}$$

Since $f(o_i) = G\, s_i$,

$$\frac{\partial f(o_i)}{\partial s_i} = G \tag{19}$$

Therefore, according to the Chain Rule,

$$\frac{\partial \mathcal{J}}{\partial \lambda} = \frac{1}{\sum_{i=1}^G |o_i|} \sum_{i=1}^G L_i \frac{\partial f(o_i)}{\partial s_i} \cdot \frac{\partial s_i}{\partial \lambda}$$

$$= \frac{G}{\sum_{i=1}^G |o_i|} \left[ \sum_{i=1}^G L_i \cdot s_i \left( \frac{\partial g_i}{\partial \lambda} - \sum_{j=1}^G s_j \frac{\partial g_j}{\partial \lambda} \right) \right]$$

Plugging in $\frac{\partial g_i}{\partial \lambda} = h_i^\lambda \ln h_i$ yields

$$\boxed{\frac{\partial \mathcal{J}}{\partial \lambda} = \frac{G}{\sum_{i=1}^{G} |o_i|} \left[ \sum_{i=1}^{G} L_i \cdot s_i \left( h_i^\lambda \ln h_i - \sum_{j=1}^{G} s_j h_j^\lambda \ln h_j \right) \right]} \tag{20}$$

## D  CHAT TEMPLATE

In this section, we present the chat template used during training. The template follows the format of role-tagged messages, with system instructions, user input, and assistant output.

**Chat Template**

```
<|im_start|>system
You are a helpful assistant.
<|im_end|>

<|im_start|>user
{input}
Please reason step by step, and put your final answer within
\boxed{{}}.
<|im_end|>

<|im_start|>assistant
```

## E  3B MODEL TRAINING CURVE

We include the training curves for the *Qwen2.5-3B* model in Figure 5 for additional insight. Overall, the 3B model shows a trend similar to the 1.5B case, with our proposed $\lambda$-GRPO consistently outperforming GRPO and DAPO on most benchmarks. DAPO shows a similar early rise (steps 0–80) but declines around steps 120–160. By contrast, GRPO improves more slowly in early steps yet catches up to DAPO near step 120. Compared to DAPO's early surge and later drop, $\lambda$-GRPO delivers steadier improvements and better late-stage accuracy on most benchmarks, while avoiding the mid-to-late degradation seen with DAPO and the slower early progress of GRPO.

## F  BENCHMARK STATISTICS

The detailed statistics of the eight benchmarks is shown in Table 2, including gsm8k, math500, minerva, gaokao, olympiad, college math, aime24, and amc23.

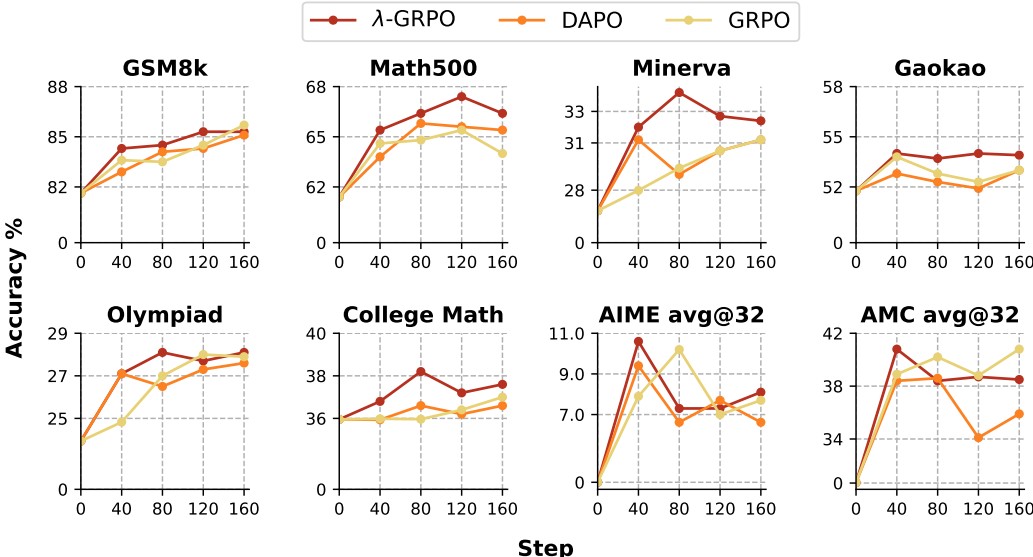

Figure 5: Training curves for the *Qwen2.5-3B* base model across benchmarks. Each panel shows accuracy (%) versus training step for three methods (*λ-GRPO*, *DAPO*, and *GRPO*). Steps are {0, 40, 80, 120, 160}.

Table 2: Different benchmarks that span diverse levels of difficulty and coverage, from grade-school word problems to Olympiad and college-level mathematics.

| Question set | Description |
|---|---|
| GSM8K | High quality linguistically diverse grade school math word problems (1319) |
| MATH500 | A representative subset from the MATH benchmark, covering challenging competition-style problems across algebra, geometry, probability, and number theory (500) |
| Minerva | A curated slice of the Minerva benchmark focusing on advanced STEM reasoning problems sourced from undergraduate-level math and science exams (272) |
| Gaokao | Math questions from Chinese college entrance examinations, requiring multi-step symbolic reasoning (385) |
| Olympiad | OlympiadBench problems, emphasizing high-school Olympiad-level creative problem solving (675) |
| College Math | Random subsamples of college-level math exam questions, selected with `random.seed(42)` for reproducibility (500) |
| AIME24 | Problems from the 2024 American Invitational Mathematics Examination, designed for top high-school students (30) |
| AMC23 | Problems from the 2023 American Mathematics Competition, targeting high-school competition-level reasoning (40) |

