# OpenReview forum: "$\lambda$-GRPO: Unifying the GRPO Frameworks with Learnable Token Preferences"
_ICLR.cc/2026/Conference — ICLR 2026 Conference Withdrawn Submission_

### Official Review · Reviewer_L9cS · 2025-10-17

**Soundness:** 1
**Presentation:** 2
**Contribution:** 2
**Rating:** 2
**Confidence:** 5

**Summary:**

The paper studies length bias in Group Relative Policy Optimization (GRPO) style RL with verifiable rewards for LLM reasoning tasks. The authors propose λ-GRPO, a unified view of GRPO-style objectives for RLVR, and introduce a learnable sample-level weighting over responses, adjusting the length preferences during policy optimization. Experiments on Qwen2.5 (1.5B/3B/7B) trained with rule-based reward show consistent gains over GRPO across math reasoning benchmarks.

**Strengths:**

1.	The proposed method is simple and can be easily dropped into existing GRPO algorithms.

2.	The comparison with GRPO, DAPO, Dr. GRPO's objective is clear.

**Weaknesses:**

1.	The introduction motivates with RLHF's reward model length bias (reward models favor longer responses) and then implies analogous issues in GRPO. However, the paper’s setting is RLVR with verifiable rewards (no reward model). The work does not theoretically or empirically demonstrate that GRPO itself induces a harmful length bias under RLVR

2.	This paper does not show or quantify the severity of length bias in GRPO. The readers are unsure if this is an important issue or would hurt learning.

3.	The converged response length in Figure 4 for the proposed λ-GRPO is similar to that of GRPO and DAPO, which suggests length bias of GRPO may have limited effect in practice.

4.	The empirical improvement over baseline methods is around 1%, which is quite marginal.

5.	All experiments use Qwen2.5 and math tasks. Experiments with other model families (Llama) should be conducted to demonstrate general RLVR applicability.

**Questions:**

Please see Weaknesses

---

### Official Review · Reviewer_UT3B · 2025-10-19

**Soundness:** 2
**Presentation:** 2
**Contribution:** 2
**Rating:** 4
**Confidence:** 3

**Summary:**

The paper proposes $\lambda$-GRPO, which introduces a learnable scalar $\lambda$ to re-weight token-level aggregation in GRPO-style objectives via a groupwise softmax over length-normalized factors . Authors claim this “lets the model learn its token preference” and report gains on 8 math benchmarks with Qwen2.5 (1.5B/3B/7B).

**Strengths:**

Presents a clean unification view for GRPO/DAPO/Dr.GRPO's token-level aggregation weight within the same clipped surrogate; $\lambda$-GRPO parameterizes this with a single trainable $\lambda$.
Simple, drop-in mechanism with minimal engineering overhead; integrates easily into existing RLVR/GRPO pipelines.

**Weaknesses:**

1. Term clarity (L191–L204): “Let the model decide its token preference” is confusing; what's the concrete definition of the "token preference". In the GRPO/DAPO algorithm, the
2. The comparison to existing methods feels incomplete. For instance, DAPO appears to have other methodological differences from GRPO beyond the length-reweighting, but these are not clearly discussed or ablated in the paper.
3. Theory gap (Method 2.3–2.4; ≈L480–L611): The paper does not sufficiently explain why existing methods, such as DR.GRPO, are unable to solve the problem of "varying length imbalancing reward" that is introduced in Section 2.3. Lacks analysis of when length-based reweighting improves bias or stability; no characterization of λ’s optimum compared to varying length–reward.
4. Scope/solidity (Results §4; Tables/Figs ≈L724–L831): Only math reasoning tasks with modest gains;  Experiment scope & solidity: Evaluations are only math reasoning on Qwen2.5 (1.5B/3B/7B) with modest average gains (+~1–2 pts). The paper lacks multi-seed variance, task diversity (code/agents/tools), and sensitivity to key parameters.
5. Granularity of “learnable” (Results org. ≈L724–L733; Table 1 ≈L789–L806): λ differs across model sizes, but not per task or per question/prompt; this undermines the “context-aware” narrative.

**Questions:**

1) Define “token preferences” precisely (≈L191–L204; L480–L611):
   What exactly is learned beyond a length-based exponent? How does it differ from fixed aggregation in GRPO/DAPO/Dr.GRPO, and why is it “context-aware”?
2) DAPO differences vs. GRPO (≈L191–L204):
DAPO is not just “uniform token averaging.” It redefines the aggregation objective to equalize per-token gradient mass across responses, decoupling sample-level rewards from token-level weights to explicitly counter length bias (e.g., per-token normalization/weight sharing, altered clipping/variance handling, token-wise advantage scaling). The paper compresses this to a “token-mean” heuristic and omits these design choices and their training-dynamics implications, making comparisons to GRPO/λ-GRPO incomplete. Question: Beyond uniform token aggregation, what other implementation differences (e.g., advantage computation/normalization across tokens or groups) are present, and how are they controlled in comparisons?
3) Why omit Dr.GRPO results? (≈L718–L721):
The claim that Dr.GRPO is “equivalent up to a constant” needs proof. Please provide a formal derivation specifying the constant and conditions; otherwise, include a Dr.GRPO baseline to substantiate the claim.
4) λ variation & granularity (Impl./Results ≈L614–L624; L724–L806):
   Please report λ trajectories/statistics per model size and justify why λ is only varied across models but not per task or per question/prompt. Would conditioning λ (e.g., by prompt features/reward noise) outperform a single global scalar?
5) Robustness:
   Provide multi-seed CIs and sensitivity to λ learning rate, group size G, given the small average gains.

---

### Official Review · Reviewer_mYft · 2025-10-31

**Soundness:** 2
**Presentation:** 2
**Contribution:** 2
**Rating:** 2
**Confidence:** 5

**Summary:**

This paper proposes λ-GRPO, a RL method incorporating a learnable token-weighting formulation for rollouts in a group. By introducing a trainable parameter λ that adaptively controls token-level weighting, λ-GRPO allows the model to learn its own length preferences during optimization. Experiments on Qwen2.5 models show improvements across multiple reasoning benchmarks.

**Strengths:**

This paper proposes a method that incorporates a length preference factor into RLVR to adaptively control length-based weighting during policy optimization. The experiments demonstrate consistent improvements over baseline methods.

**Weaknesses:**

1. The presentation of the “DAPO” method in this paper is rather informal. Although the authors mention that the “DAPO” used here is a variant of GRPO under their specific setup (line 305), the original DAPO method [1] is already a widely recognized RLVR approach. This reuse of the name may cause confusion for readers. Moreover, some well-established strategies in DAPO, such as Clip-Higher, have been shown effective. However, the paper only compares its approach against the original weaker version of GRPO, which weakens the overall persuasiveness of its results.
   Furthermore, in line 420, the authors claim that DAPO exhibits higher entropy, but this effect is due to the Clip-Higher mechanism, which they did not include in their implementation. More concerningly, they still use their so-called “DAPO” to support claims about entropy maintenance, which constitutes a serious methodological error.


2. The results for the base models, particularly Qwen2.5-1.5B, appear weaker than expected. According to the original Qwen2.5 technical report [2], the Qwen2.5-1.5B model achieves 68.5 on GSM8K and 35.0 on MATH, whereas this paper reports only 8.5 and 1.8, respectively. Although the technical report evaluates models under few-shot settings, the performance reported here is still implausibly low based on prior practical experience [3].

3. The method is proposed for training length preferences in RL; however, the intermediate response lengths appear nearly identical to those in the so-called “DAPO” setting shown in Figure 4(b), which is effectively equivalent to fixing the λ value at 1. Providing the trends or analyses of intermediate λ values during training, or illustrating specific cases that highlight the role of λ, could make the method more interpretable.

[1] Dapo: An open-source llm reinforcement learning system at scale

[2] Qwen2.5 Technical Report

[3] SimpleRL-Zoo: Investigating and Taming Zero Reinforcement Learning for Open Base Models in the Wild

**Questions:**

1. The authors stated that Dr. GRPO is merely a rescaling of DAPO and therefore chose not to include it in their comparison (line 307). However, in Equation (16) (line 237), their own method also applies a scaling operation to the total weight of the length-controlling factor. Could the authors clarify how this scaling differs from the one used in Dr. GRPO?

---

### Official Review · Reviewer_rqYu · 2025-10-31

**Soundness:** 3
**Presentation:** 1
**Contribution:** 2
**Rating:** 4
**Confidence:** 4

**Summary:**

The paper introduces λ-GRPO, a loss aggregation variant of GRPO designed to mitigate the length-bias issue in LLM RL post-training. By unifying the loss aggregation formulation of GRPO and DAPO, the authors introduce a learnable λ parameter that enables adaptive control over response length preference. λ-GRPO achieves stronger performance on math reasoning benchmarks than both GRPO and DAPO.

**Strengths:**

1. The paper provides a clear summary of the differences between GRPO and DAPO in their loss aggregation mechanisms.

2. The empirical results on mathematical reasoning tasks are relatively strong, demonstrating a non-negligible performance improvement.

**Weaknesses:**

### Methodology

1. The modification to the loss weighting—both its formulation and the introduction of the learnable parameter λ—feels overly ad-hoc. The authors claim that this alleviates the need to “fix them by hand as in prior approaches,” but in my view, the proposed formulation appears even more hand-crafted than those in GRPO or DAPO.

2. The weighting term \((1 + r z)^{\lambda}\) lacks theoretical justification and seems potentially problematic. A **linear approximation** would suffice. Although the authors assume that response lengths follow a Gaussian distribution (an assumption that is questionable) and thus argue that \(1 + r z\) stays near 1 with high probability, they then adopt a rather “aggressive” setting of \(r = 1/9\), increasing the likelihood that \(1 + r z\) deviates from 1. Once \(1 + r z \le 0\) (i.e., \(r z \le -1\)), the expression becomes undefined. If the authors believe such large deviations are negligible, they should admit that the safe magnitude of \(r z\) is indeed much smaller than 1, in which case, as long as λ is not excessively large, the term could be **well approximated by \(1 + r \lambda z\)**. This linearized form would still capture the qualitative role of λ described in the paper, unless the authors can demonstrate that the learned λ induces substantial nonlinear effects in the exponential term—which they do not.

---

### Experiments and Result Analysis

1. The experiments are restricted to **mathematical reasoning tasks only**.
2. The paper severely lacks empirical insight and in-depth analysis supporting its methodological claims. While performance gains are observed on specific tasks, the authors do not show how the learnable λ actually evolves during training or how it shapes response-length preferences in practice. The length-bias motivation alone does not sufficiently justify the proposed design.
3. The entropy analysis shows no significant difference between λ-GRPO and DAPO. Moreover, GRPO—highly relevant for length comparison—is omitted from this analysis, making the entropy section appear more like trend-following than substantive investigation.
4. The response-length analysis similarly shows **no clear difference** between λ-GRPO and DAPO, casting further doubt on the empirical validity of the claimed length-bias mitigation. A constant-weight baseline such as DAPO already seems adequate.

---

### Writing and Presentation

1. Several notations are missing or inconsistently used. For example, in lines 209–210, the paper uses undefined symbols *o* in *mean(o)* and *std(o)*, and in line 213, subtracts μ from *oᵢ* (where *oᵢ* is defined as an element in the response set. You cannot subtract a scalar value directly from a response.). This inconsistency reduces the paper’s professionalism and readability.
2. The paper repeatedly discusses **Dr.GRPO** and frequently mentions improvements in DAPO and Dr.GRPO unrelated to loss aggregation, which adds verbosity without contributing to the core methodological or empirical narrative.
3. The highlighted table indicating that λ is initialized as a `torch.tensor` is unnecessary.

**Questions:**

See weaknesses.

---

### Note · Authors · 2025-11-19

I have read and agree with the venue's withdrawal policy on behalf of myself and my co-authors.